# Impact and Cost-Effectiveness of Alternative Human Papillomavirus Vaccines for Preadolescent Girls in Mozambique: A Modelling Study

**DOI:** 10.3390/vaccines11061058

**Published:** 2023-06-02

**Authors:** Esperança Lourenço Guimarães, Assucênio Chissaque, Clint Pecenka, Frédéric Debellut, Anne Schuind, Basília Vaz, Arlindo Banze, Ricardina Rangeiro, Arlete Mariano, Cesaltina Lorenzoni, Carla Carrilho, Maria do Rosário Oliveira Martins, Nilsa de Deus, Andrew Clark

**Affiliations:** 1Instituto Nacional de Saúde, Marracuene District, EN1, Bairro da Vila—Parcela N° 3943, Maputo 1120, Mozambique; 2Global Health and Tropical Medicine, Instituto de Higiene e Medicina Tropical (IHMT), Universidade Nova de Lisboa, Junqueira Street 100, 1349-008 Lisbon, Portugal; 3Center for Vaccine Innovation and Access, PATH, Seattle, WA 98121, USA; 4Center for Vaccine Innovation and Access, PATH, 1202 Geneva, Switzerland; 5Ministry of Health, Maputo 1008, Mozambique; 6National Cancer Control Program, Hospital Central de Maputo, Maputo 1101, Mozambique; 7Department of Pathology, Universidade Eduardo Mondlane, Maputo 3453, Mozambique; 8Department of Health Services Research and Policy, London School of Hygiene & Tropical Medicine, London WC1E 7HT, UK

**Keywords:** cervical cancer, papillomavirus, vaccination, modelling, UNIVAC, cost-effectiveness, Mozambique

## Abstract

Mozambique has one of the highest rates of cervical cancer in the world. Human papillomavirus (HPV) vaccination was introduced in 2021. This study evaluated the health and economic impact of the current HPV vaccine (GARDASIL^®^ hereafter referred to as GARDASIL-4) and two other vaccines (CECOLIN^®^ and CERVARIX^®^) that could be used in the future. A static cohort model was used to estimate the costs and benefits of vaccinating girls in Mozambique over the period 2022–2031. The primary outcome measure was the incremental cost per disability-adjusted life-year averted from a government perspective. We conducted deterministic and probabilistic sensitivity analyses. Without cross-protection, all three vaccines averted approximately 54% cervical cancer cases and deaths. With cross-protection, CERVARIX averted 70% of cases and deaths. Without Gavi support, the discounted vaccine program costs ranged from 60 million to 81 million USD. Vaccine program costs were approximately 37 million USD for all vaccines with Gavi support. Without cross-protection, CECOLIN was dominant, being cost-effective with or without Gavi support. With cross-protection and Gavi support, CERVARIX was dominant and cost-saving. With cross-protection and no Gavi support, CECOLIN had the most favorable cost-effectiveness ratio. Conclusions: At a willingness-to-pay (WTP) threshold set at 35% of Gross Domestic Product (GDP) per capita, HPV vaccination is cost-effective in Mozambique. The optimal vaccine choice depends on cross-protection assumptions.

## 1. Background

Cervical cancer is the fourth most diagnosed cancer among women globally, with an estimated 342,000 deaths worldwide in 2020 [1,2]. Around one in five cervical cancer deaths are estimated to occur in sub-Saharan Africa [1,3]. Mozambique had the eighth highest age-standardised cervical cancer mortality rate (38.7 per 100,000 women) in the world, with 5325 new cases and 3850 deaths, in 2020 [4]. This accounted for 21.4% of all female cancer deaths in the country.

Cervical cancer is caused by persistent high-risk human papillomavirus (HPV) infection, which is mainly spread through sexual contact [5]. In Mozambique, cervical HPV infections have been identified in almost two-thirds of women aged 18–24 years. Screening for pre-cancerous lesions and vaccination against HPV infection prior to sexual debut are safe and effective ways to prevent cervical cancer [6,7]. In 2009, Mozambique started the National Screening Program for Cervical Cancer, using the visual inspection with 3-4% acetic acid (VIA) method, targeting women aged 30–55 years of age every five years [8,9,10,11]. However, in 2014/2015, the self-reported coverage of cervical cancer screening uptake using cytology and VIA among women aged 30–55 years was estimated to be only 3.5% [8]. From April 2018 to September 2019, Mozambique performed a hospital-based pilot screening demonstration project using primary HPV DNA testing, but it was never implemented at the national level [12].

In addition to cervical cancer screening at older ages, HPV vaccination is the primary prevention measure that can benefit pre-adolescents or adolescents. HPV vaccines have now been introduced in over 100 countries worldwide [13], with several studies demonstrating favorable cost-effectiveness [6,14,15]. Four vaccines are currently pre-qualified by the World Health Organization (WHO): CECOLIN^®^, CERVARIX^®^, GARDASIL^®^ (referred to hereafter as GARDASIL^®^-4), and GARDASIL^®^-9. All four vaccines cover HPV genotypes 16 and 18. GARDASIL-4 covers additional two types (6 and 11) that are responsible for anogenital warts. GARDASIL-9 covers an additional seven types (6, 11, 31, 33, 45, 52, and 58), but is not eligible for external funding from Gavi, the Vaccine Alliance (Gavi). All vaccines were indicated to be administered in two doses given six months apart to pre-adolescent girls aged 9–14 years [16,17]. However, the WHO Strategic Advisory Group of Experts on Immunization (SAGE) recently recommended a single-dose schedule, since it provides similar efficacy to the two or three-dose regimens [18,19].

In 2014 and 2015, Mozambique performed a school-based HPV vaccine demonstration project in the districts of Manhiça, Manica, and Mocímboa da praia using CERVARIX [20]. In November 2021, with support from Gavi, the country introduced GARDASIL-4 into the national program on immunization. In the first phase, the vaccine was administered to girls aged 9 years via community outreach brigades (28%), health centres (22%), and schools (50%) [21]. 

An economic evaluation is required to assess the health and economic impact of the recently introduced GARDASIL-4 vaccine and the cost-effectiveness of two alternative vaccines that are currently eligible for funding by Gavi (CECOLIN and CERVARIX). This also provides an opportunity to explore the potential costs and benefits of different strategies, e.g., a single-dose schedule or multi-age-cohort [MAC] campaign. This will provide important evidence to decision-makers about the value for money of the current HPV vaccine and alternative products and strategies that could be used in the future.

## 2. Methods

### 2.1. Study Design

From a government perspective, we evaluated the cost-effectiveness of three HPV vaccines (CECOLIN, CERVARIX, and GARDASIL-4), each compared to no vaccination (and no change in screening practices) and to each other. In our base case scenario, we evaluated the lifetime costs and benefits of vaccinating nine annual cohorts of 9-year-old girls (routine vaccination 2022–2031) and five cohorts of girls aged 10–14 years (catch-up MAC campaign conducted in the year 2022) at the national level.

A multidisciplinary group of experts was invited to a stakeholder consultation workshop (10–11 May 2022) to provide feedback on the inputs and assumptions used in the analysis. This included stakeholder representatives from the Expanded Program of Immunization (EPI), the National Immunization Technical and Advisory Group (known in Mozambique as Comité de Peritos de Imunização), the National Cancer Control Program, WHO, and the United Nations Children’s Fund.

### 2.2. Modelling Approach

We used the UNIVAC decision-support model (version 1.54), an Excel proportionate outcomes static cohort model [22]. UNIVAC is populated with the United Nations (2019 revision) population estimates of the number of girls alive in each year and calendar year of life over the lifetimes of all birth cohorts included in the analysis [23]. Numbers of girls alive in each single year/age of life are multiplied by age-specific rates of cervical cancer cases and deaths to estimate the number of cases, deaths, and disability-adjusted life years (DALYs) expected to occur with and without vaccination over the lifetimes of each cohort of vaccinated girls. The model also estimates the costs of vaccination and the healthcare costs associated with treating cervical cancer cases, with and without vaccination.

In addition, the model requires estimates of age-specific cervical cancer incidence and mortality by stage, rates of access to healthcare and associated treatment costs, vaccination program costs, and the expected coverage and efficacy of each vaccine.

The primary outcome measure was the cost (2021 USD) per DALY averted from a government perspective. DALYs were used because they combine both years lost due to premature death and years lived with disease and allow health effects to be compared consistently across diseases. Future health and cost outcomes were discounted at 3% per year to reflect the time preference for immediate benefits and the opportunity of investing present capital [24].

Mozambique has not yet defined a country-specific WTP threshold to determine whether an intervention is cost-effective. A previous study recommended that countries with a low human development index should use a threshold below 100% of the GDP per capita based on the revealed WTP of many low- and middle -income countries [25]. However, for Mozambique, others studies have recommended a threshold of 16 to 35% of the GDP per capita [26]. Given the uncertainty around this threshold, we calculated the probability that HPV vaccination would be cost-effective for WTP thresholds ranging from 0% to 35% of the GDP per capita. This is equivalent to USD 175 based on a national GDP per capita of USD 500 in November 2022 [27].

### 2.3. Disease Burden

Input data for disease burden are summarized in Table 1. We used age-specific rates of cervical cancer cases and deaths estimated for Mozambique by Globocan for the year 2020 and assumed these rates would remain constant over time in the absence of vaccination or any changes to current screening practices [28]. We assumed cases were distributed into 18.6% local, 72.9% regional, and 8.5% distant cervical cancer, based on a cancer stage distribution previously estimated for countries in the low-income/lower–middle-income strata [29]. Cancer stage definitions are based on the surveillance, epidemiology, and end results (SEER), as well as the International Federation of Gynecology and Obstetrics (FIGO) staging system [30]. Disability weights represent time lost, while living with local, regional, and distant cancer were taken from the Global Burden of Disease project [31].

The percentage of women alive five years after diagnosis was estimated for each stage based on the data from a recent study of survival rates from several sub-Saharan African countries [32]. In this study, Mozambique’s three-year survival percentage from all stages was very similar to Kenya’s (both around 55%). Data on five-year survival was not reported for Mozambique, so we assumed the five-year survival reported for Kenya (44%) [32]. To estimate survival by stage (local, regional, and distant), we applied the ratio between all-stage survival and stage-specific survival, as recently reported in the United States of America [30]. The resulting five-year survival rates (61%, 39%, and 12% for local, regional, and distant cervical cancer, respectively) are broadly consistent with estimates for the low human development index [32]. For all parameters without uncertainty ranges, we varied the central estimate by±20% to generate a plausible range for use in uncertainty analysis [33,34].

**Table 1 vaccines-11-01058-t001:** Input parameters for estimating cervical cancer disease burden.

Parameter	Base Case	Uncertainty Range		Source
		Low	High	
Age-specific rates, 100,000 per year, cervical cancer cases				
10–15	0.6	0.4	0.7	[28]
15–20	2.7	2.2	3.3	
20–25	17.8	14.3	21.5	
25–30	34.9	27.9	41.9	
30–35	55.1	44.1	66.1	
35–40	76.5	61.3	91.9	
40–45	99.8	79.8	119.8	
45–50	120	96.1	144.1	
50–55	131.5	105.3	157.7	
55–60	136.5	109.2	163.9	
60–65	130.7	104.7	157.0	
65–70	120.6	96.4	144.7	
70–75	107.6	86.0	129.0	
75–80	92	73.6	110.4	
80–85	72.1	57.6	86.6	
85–90	53.5	42.8	64.2	
90–95	53.5	42.8	64.2	
95–100	53.5	42.8	64.2	
Percentage of cervical cancer cases in each stage				
Local cancer ^a^	18.6	17.9	22.3	[29]
Regional cancer ^b^	72.9	70.0	87.5	
Distant cancer ^c^	8.5	8.2	10.2	
Age-specific rates, 100,000 per year, cervical cancer deaths				
10–15	0.4	0.3	0.5	[28]
15–20	2.8	2.2	3.4	
20–25	8.3	6.6	10.0	
25–30	16.8	13.4	20.2	
30–35	28.7	23.0	34.4	
35–40	43.6	34.9	52.3	
40–45	65.3	52.2	78.4	
45–50	87.2	69.8	104.6	
50–55	107.2	85.8	128.6	
55–60	122.9	98.3	147.5	
60–65	126.4	101.1	151.7	
65–70	123.3	98.6	148.0	
70–75	113.1	90.5	135.7	
75–80	96.8	77.4	116.2	
80–85	75.9	60.7	91.1	
85–90	48.7	39.0	58.4	
90–95	48.7	39.0	58.4	
95–100	48.7	39.0	58.4	
Percentage of healthy time lost while living with disease				
Local cancer ^a^	28.8	19.3	39.9	[31]
Regional cancer ^b^	45.1	30.7	60.0	
Distant cancer ^c^	54.0	37.7	68.7	
Average 5-year survival rate (% alive after 5 years)				
Local cancer	60.7	72.8	48.6	Assumed based on [32,35]
Regional cancer	38.3	45.9	30.6	
Distant cancer	11.9	14.3	9.5	

^a^ Local cancer refers to FIGO stage 1 and 2. ^b^ Regional cancer refers to FIGO stage 3. ^c^ Distant cancer refers to FIGO stage 4.

### 2.4. Health Service Utilization and Costs

We assumed that all women captured in the Globocan incidence rates would be diagnosed and that 91% of these women would go on to receive treatment [36]. Therefore, estimates of the average cost of cervical cancer treatment were only applied to women who were both diagnosed and treated. Since there are no data on cervical cancer treatment costs in Mozambique, we used data from a cost of illness study performed in Tanzania from a government perspective. This included direct medical costs for labor, supplies, equipment, and patient hospital accommodation/admission (Table 2) [37]. These costs were originally in 2013 USD, so we converted them to 2021 USD [38].

### 2.5. Vaccine Coverage and Efficacy

Vaccine inputs on vaccine coverage and efficacy are presented in Table 3. The first and second dose coverages were estimated to be 93% and 17%, respectively, for 2022, based on coverage reported by the EPI program for the current HPV vaccine [21]. For the period of 2023–2031, we assumed 93% and 73%, respectively, based on measles coverage in children aged 10–14 years in 2018 [39]. The same coverage (93 and 73%) was assumed for the catch-up campaign in the first year.

Some studies have indicated potential cross-protection against HPV genotypes not covered by the vaccines [40,41]. However, there is uncertainty about how much cross-protection should be assumed for each vaccine. We, therefore, modelled the cost-effectiveness of each vaccine with and without cross-protection. The efficacy of the complete (two dose) vaccination schedule was taken from clinical trials of efficacy against high-grade lesions, i.e., cervical intra-epithelial neoplasia [42,43,44]. An overall weighted efficacy value was calculated by multiplying the efficacy assumed for each HPV type by the proportion of cervical cancers caused by each type in Mozambique. The type distribution for Mozambique was taken from the HPV Information Centre. The top three HPV types in Mozambique were 18 (43.0%), 16 (20.4%), and 45 (11.9%) [9]. The overall weighted efficacies of CECOLIN, CERVARIX, and GARDASIL-4 were estimated to be 63% [42], 63% [43,45,46], and 62% [47,48], respectively, without cross-protection, and 64% [42], 83% [43,45,46], and 63% [47,48], respectively, with cross-protection. The influential cross-protection assumptions for CERVARIX were taken from the study by Wheeler et al. [43]. For all vaccines, we multiplied the two-dose efficacy values by 0.8 (range 0.7–1.0) to estimate the efficacy of one dose.

**Table 3 vaccines-11-01058-t003:** Input parameters for estimating the health impact of HPV vaccination.

Parameter	Value	Uncertainty Range		Source/s
		Low	High	
Coverage for routine vaccination and catch-up campaign (2022–2031)				
1st dose (2022–2031)	93.0%	74.0%	98.0%	[21]
2nd dose (2022)	17.0%	13.6%	20.4%	
2nd dose (2023–2031)	73.0%	58.4%	87.6%	[39]
Vaccine efficacy (all types combined) with cross-protection				
CECOLIN				
Dose 1	51.4%	30.2%	51.6%	Assumption (80% of 2 doses)
Dose 2	64.3%	37.8%	64.5%	[42]
CERVARIX				
Dose 1	66.1%	48.3%	67.9%	Assumption (80% of 2 doses)
Dose 2	82.7%	60.3%	84.9%	[43,45,46]
GARDASIL-4				
Dose 1	50.4%	45.4%	51.3%	Assumption (80% of 2 doses)
Dose 2	63.0%	56.7%	64.1%	[47,48]
Vaccine efficacy (all types combined) without cross-protection				
CECOLIN				
Dose 1	50.7%	29.9%	50.7%	Assumption (80% of 2 doses)
Dose 2	63.4%	37.4%	63.4%	[42]
CERVARIX				
Dose 1	50.1%	40.7%	50.7%	Assumption (80% of 2 doses)
Dose 2	62.7%	50.8%	63.4%	[43,45,46]
GARDASIL-4				
Dose 1	49.7%	45.0%	50.4%	Assumption (80% of 2 doses)
Dose 2	62.1%	56.3%	63.0%	[47,48]

Note: We have assumed a type distribution based on Information Centre on HPV and Cancer. We assume lifelong protection from vaccination. Cross protective efficacy was assumed against HPV types 31, 33, 45, 51, 52 and 56 for CERVARIX [43,45,46], and against type 31 for GARDASIL-4 [47,48]. We further assumed the same cross-protection against type 31 for CECOLIN.

### 2.6. Vaccination Program Costs

Mozambique is currently eligible for vaccine financial support from Gavi. This means the majority of the manufacturer’s vaccine price is paid for by Gavi, and only USD 0.20 per dose is paid for by the government [49]. However, we also presented our results without Gavi support to show how the cost-effectiveness would be impacted if the government were to pay the full manufacturer’s vaccine price over the full ten-year period. We assumed that the prices would be fixed for the entire period of the analysis. In both scenarios (with and without Gavi support), the government is expected to cover all the costs associated with wastage, procurement related charges, and integrating the vaccine into the current immunization program. In the first year of vaccine implementation, Gavi provides a vaccine introduction grant (VIG) equivalent to USD 2.40 per girl aged 9 years (USD 1,327,040) and USD 0.65 per girl aged 10–14 years (USD 1,793,700), and all MAC vaccines (10–14 years) are provided at no cost [50].

HPV vaccination program costs were calculated by combining the United Nations estimates of the number of girls in the target ages/years with estimates of HPV vaccine coverage and the input data presented in Table 4, namely the vaccine price, wastage, international handling (procurement process), international delivery, and immunization delivery cost (which comprises the additional cost to the health system that would be involved from adding the vaccine to the current vaccine delivery system and representing expenses related to supply chain, capital, labor, and other service delivery costs to implement the vaccination in the country).

### 2.7. Uncertainty Analysis

We ran univariate (one-way) deterministic scenario analyses to estimate the influence of several model assumptions and input values on the cost-effectiveness results [24]. One scenario evaluated the cost-effectiveness of a single dose of HPV vaccination (with full protection assumed for one single dose), consistent with a recent study from Kenya [10]. We ran one additional scenario for CERVARIX (with cross-protection) excluding any cross-protective benefits for types HPV-52 and HPV-56 because a study by Wheeler et al [43] has suggested any reported health benefit for these types might be due to chance observations. Other scenarios unfavourable to vaccination included low vaccine coverage, low average treatment costs, discount rate at 10%, low disease burden, and no MAC. Scenarios favourable to vaccination included high vaccine coverage, high average treatment costs, and high disease burden. In addition, we ran a probabilistic sensitivity analysis (PSA), varying all parameters simultaneously within their uncertainty ranges, assuming simple BETA-Pert distributions for each parameter [53]. Prices were assumed to be fixed within the PSA. We ran separate PSAs for each vaccine, with and without cross-protection, with 1000 runs per vaccine/scenario. PSA results were presented on a cost-effectiveness plane and used to estimate the probability that each vaccine would be cost-effective at different WTP thresholds.

## 3. Results

### 3.1. Base Case Analyses

Without HPV vaccination in Mozambique, we estimate there could be 342,246 cases of cervical cancer, 282,687 deaths, and 1,695,103 DALYs lost over the lifetimes of 14 cohorts of preadolescent girls (Table 5).

With Gavi support, each of the three vaccines would cost around USD 37 million (USD 42 million undiscounted), compared to no vaccination (Table 5 and Table 6). Without Gavi support, vaccine program costs are estimated to be USD 60 million for CECOLIN, USD 73 million for GARDASIL-4, and USD 81 million for CERVARIX (Table 7).

In scenarios without cross-protection, all three vaccines had similar health benefits (54% reduction in cervical cancer cases and deaths) and net costs, compared to no vaccination. CECOLIN had the lowest net cost and the highest estimated impact, averting 184,669 cases, 152,528 deaths, and 908,898 DALYs (Table 5). CECOLIN, therefore, dominated both GARDASIL-4 and CERVARIX. However, subtle changes in cost and efficacy assumptions could easily change the rank order. With Gavi support, CECOLIN was the most cost-effective (USD 2.5 per DALY averted). Without Gavi support, CECOLIN was still dominant and very cost-effective (cost per DALY averted equivalent to 5% of the GDP per capita).

In scenarios with cross-protection, CERVARIX had substantially more health benefits than the other two products (70% reduction in cervical cancer cases and deaths) (Table 6). With Gavi support, CERVARIX was dominant and cost-saving (Figure 1). Without Gavi support, CECOLIN was the most cost-effective product, but CERVARIX still had very favorable cost-effectiveness; the incremental cost-effectiveness of using CERVARIX (compared directly to CECOLIN, rather than no vaccination) was USD 6, equivalent to 1% of the GDP per capita (Table 8 and Figure 2), despite CERVARIX having a substantially higher vaccine program costs than CECOLIN (81 million USD versus 60 million USD).

### 3.2. Uncertainty Analysis

One-way deterministic sensitivity analysis showed HPV vaccination with Gavi support was still cost-effective in the most unfavourable scenarios, such as higher vaccine price (no Gavi support), low disease burden rates, low vaccine coverage, low average treatment costs, and no MAC. Removing cross-protective benefits of CERVARIX against HPV-52 and HPV-56 decreased the overall weighted efficacy of CERVARIX from 83% to 81% and therefore had a minimal influence on our overall estimates of vaccine impact (70% to 69% reduction in cervical cancer cases and deaths) and cost-effectiveness. However, a very high discount rate (10%) would be influential and may change the conclusions. Under this scenario, the cost per DALY averted was equivalent to 90% (USD 448), 109% (USD 544), and 92% (USD 459) for the GDP per capita for CECOLIN, CERVARIX, and GARDASIL-4, respectively. On the other hand, none of the favourable scenarios influenced the results (Appendix A). 

Without Gavi support, there is a 100% probability that the most cost-effective vaccine will be cost-effective at a WTP threshold set at 35% GDP per capita (USD 175). Without Gavi support, and assuming cross-protection, there was a 100% probability that CECOLIN would be cost-effective at a WTP threshold of USD 175. However, comparing CERVARIX directly to CECOLIN had a similar probability of being cost-effective in this scenario (Figure 2).

In all scenarios assuming a single-dose (assuming the same efficacy as a full dose scheme), the vaccine costs were reduced substantially (Appendix A) and all products were cost-saving, compared to no vaccination, even without cross-protection or Gavi support (Appendix A).

## 4. Discussion

We evaluated the lifetime cost-effectiveness of vaccinating girls 9 years of age over the period 2022–2031 with a catch-up campaign for girls aged 10–14 years in the first year. Our findings suggest that HPV vaccination could reduce the burden of cervical cancer cases and deaths by 70–53%, depending on assumptions about cross-protection. Irrespective of the scenario (e.g., with and without cross-protection, with and without Gavi support), we find that the most cost-effective vaccine would be either cost-saving or cost-effective at a WTP threshold set at 35% GDP p.c. Others have recommended a threshold of 16–35% for Mozambique, which indicates that all of our main scenarios, even those without Gavi support or cross-protection, could represent good value for the money. A similar threshold (40%) was recently used to assess the cost-effectiveness HPV vaccination in Ghana [54]. In the deterministic sensitivity analysis, we analysed the cost-effectiveness of vaccinations with one dose schedule, assuming the same efficacy as a full dose scheme, as observed in a Kenyan study and, unsurprisingly, found this was more cost-effective and less costly than using two dose schedule. Only a discount rate of 10% generated a cost-effectiveness ratio exceeding 35% of the GDP per capita. Our results were particularly sensitive to the choice of discount rate because the benefits of HPV vaccination occur many years in the future. Assigning a higher discount rate (lower value to distant events) is, therefore, unfavourable to HPV vaccination.

Mozambique introduced GARDASIL-4 in November 2021. While our analysis suggests that HPV vaccination is likely to be good value for the money, it also suggests that different products could be considered to reduce costs and/or increase health benefits. Our analysis of the optimal choice of HPV vaccine depends on influential assumptions about cross-protection and does not incorporate the benefits of GARDASIL-4 on genital warts (non-malignant) or the switching costs that would be required to replace it with either of the two alternative products. However, under scenarios of cross-protection, we find that CERVARIX could have more impact than GARDASIL-4 and is worth consideration while both vaccines are heavily subsidized by Gavi. This is despite the higher wastage that may be associated with the CERVARIX vaccine´s presentation (considering the multi versus single dose vials) [16]. Some studies have reported the impact of CERVARIX on HPV oncogenic types other than 16 and 18, demonstrating its cross-protection potential. Kavanagh and others found that, seven years after girls vaccination in Scotland, there was a decline in vaccine and cross-protective types, namely HPV 31/33/45 [41]. In addition, with data from Papillomavirus surveillance in the Netherlands, Hoes et al. showed significant reduction in cross—protective types HPV-31/45 in women and heterosexual men [40]. In contrast to other HPV vaccines using aluminum-based adjuvants, CERVARIX uses the adjuvant AS04, a combination of the traditional adjuvant alum plus the TLR4 agonist monophosphoryl lipid A, and this may enhance the immune responses [55]. If Mozambique should graduate from Gavi support, then CECOLIN should also be considered on the basis of its low cost, relative to the other two vaccines, particularly if there is uncertainty or controversy about the relative cross-protection associated with the different products.

Beyond cost-effectiveness, there are other relevant aspects, such as affordability, sustainability, acceptability, and feasibility for the government, which should be discussed and contextualized [56,57,58]. In the absence of Gavi support, the government would need to pay the full price for the vaccine, leading to a less affordable vaccination program. Under this scenario, vaccination with CERVARIX would be the most expensive option (81 million USD), followed by GARDASIL-4 (73 million USD) and, finally, CECOLIN (60 million USD). With base case coverage assumptions, this is equivalent to undiscounted annual costs of 9 million USD, 8 million USD, and 6 million USD, respectively.

The WHO target for cervical cancer eradication is to fully vaccinate 90% of girls up to 15 years old, screen 70% of women at age 35, and again at age 45 years old, and treat 90% of diagnosed women [59]. However, in Mozambique, the only indicator currently being reached (according to a study performed in Maputo city) is the percentage of women with pre-invasive/invasive cervical disease receiving treatment (90%). However, the treatment rate in this study may not be representative of the national situation [36]. All other goals are far below the current WHO targets. Although the coverage of the first dose of HPV vaccination in 2021 was 93%, the second was only 17%, probably due to the recent introduction [21]. Furthermore, according to a national level survey, only 3.5% of the Mozambican women are screened for cervical cancer, most likely due to the lower coverage of the health service provision, lack of formal education, and low income [8]. This reinforces the need for increasing investments in health education and access to screening, to ensure socio-economic returns of the vaccination at mid-to-long-term.

Our study had a number of limitations. First, UNIVAC is a static cohort model and, therefore, excludes any potential indirect ‘herd immunity’ benefits of vaccination. However, these effects would only have made our results more favourable to vaccination. Second, we had limited country-specific information for some parameters and had to agree on reasonable inputs from alternative sources with the support of a national team of experts during a stakeholder consultation workshop. Third, we excluded costs borne by households, such as out-of-pocket medical expenses, travel, and lost earnings. However, these costs are likely to be relatively small, and a preliminary analysis with these costs included did not alter the cost-effectiveness results.

## 5. Conclusions

HPV vaccination is a cost-effective intervention in Mozambique. The optimal choice of vaccine depends on influential assumptions about cross-protection. A single-dose vaccine schedule could provide similar health benefits to two doses and may be an important way to reduce costs. The cost-effectiveness of the vaccines should be continually re-evaluated as more information emerges about their efficacy and costs.

## Figures and Tables

**Figure 1 vaccines-11-01058-f001:**
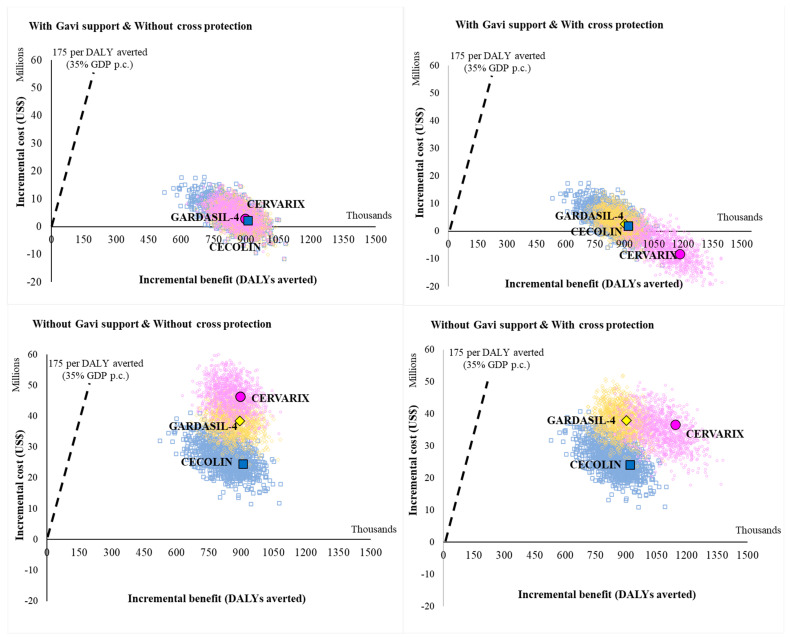
Cost-effectiveness plane showing the incremental costs and benefits of vaccination with CECOLIN, CERVARIX, and GARDASIL-4, considering Gavi support, cross-protection, and no cross-protection, compared to no vaccination.

**Figure 2 vaccines-11-01058-f002:**
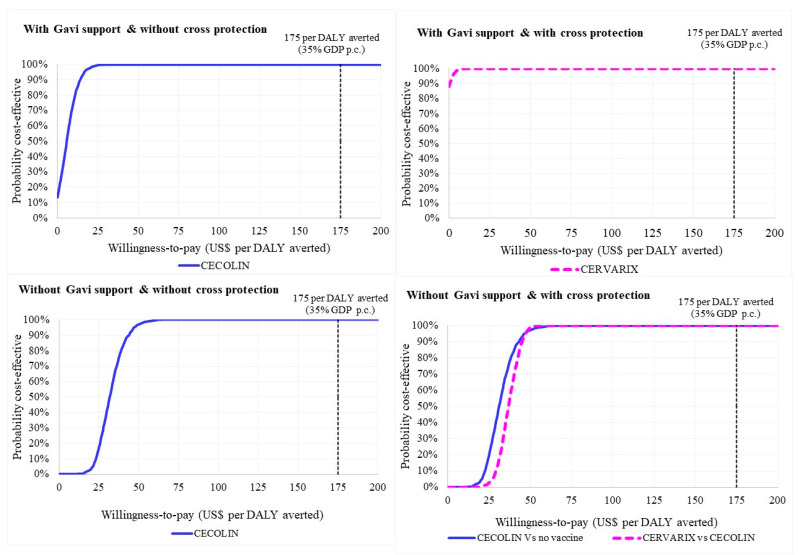
Cost-effectiveness acceptability curve for the vaccine with the most favorable cost-effectiveness ratio under different scenarios, over the period 2022–2031. Note: The first panel (top left) shows that with Gavi support and cross-protection all probabilistic sensitivity analysis runs reported a cost-saving result for product with the most favorable cost-effectiveness (CERVARIX), i.e., 100% probability of being cost-effective across all willingness-to-pay thresholds.

**Table 2 vaccines-11-01058-t002:** Input parameters for estimating health service costs from the government perspective (2021 USD).

Parameter	Base Case	Uncertainty Range		Source
		Low	High	
Local cancer				
% of diagnosed receiving treatment	91%	86%	96%	[36]
Cost per treated woman ^a^	1,188	950	1425	[37]
Regional cancer				
% of diagnosed receiving treatment	91%	86%	96%	[36]
Cost per treated woman ^b^	692	553	830	[37]
Distant cancer				
% of diagnosed receiving treatment	91%	86%	96%	[36]
Cost per treated woman ^c^	691	553	829	[37]

^a^ Local cancer refers to FIGO stage 1 and 2, of which treatment includes curative radiotherapy, chemotherapy, and surgery. ^b^ Regional cancer refers to FIGO stage 3, which treatment includes palliative radiotherapy only. ^c^ Distant cancer refers to FIGO stage 4, which treatment includes palliative radiotherapy only.

**Table 4 vaccines-11-01058-t004:** Vaccination cost inputs.

Parameter	Base Case	Uncertainty Range		Source
		Low	High	
Price of vaccine doses (USD)				
CECOLIN	0.20	-	2.90 *	[16]
CERVARIX	0.20	-	5.18 *	
GARDASIL-4	0.20	-	4.50 *	
Handling and delivery (% of price)				
% International handling	3.0	2.4	3.60	[51]
% International delivery	10.0	8.0	12.0	
Wastage percentage (%)				
CECOLIN	5.0	3.8	6.3	[16]
CERVARIX **	10.0	3.8	6.3	
GARDASIL-4	5.0	3.8	6.3	
Other costs				
Syringe price per dose (USD)	0.05	0.04	0.06	[16]
Syringe percentage wastage (%)	10.0	8.0	12.0	[20]
Costs of safety box per dose (USD)	0.01	0.01	0.01	[16]
Incremental Cost for delivery (USD) ***				
Costs per dose (2023–2031))	3.76	3.0	4.5	[52]

* Vaccine price without Gavi financing. ** CERVARIX wastage was assumed to be higher than the other vaccines, due to a multi-dose vial presentation. *** Includes Cold chain, planning and training, social mobilization, supervision, and service delivery, which was the biggest ingredient accounting for 24% of the total incremental delivery cost.

**Table 5 vaccines-11-01058-t005:** Lifetime effects and costs of vaccinating 14 cohorts of preadolescent girls over the period 2022–2031 in Mozambique (with Gavi support, without cross-protection).

OUTCOMES	No Vaccine	CECOLIN	GARDASIL-4	CERVARIX
HEALTH OUTCOMES				
Cervical cancer cases (local)	63,637	29,299	29,527	29,719
Cervical cancer cases (regional)	249,451	114,852	115,744	116,496
Cervical cancer cases (distant)	29,158	13,425	13,529	13,617
Cervical cancer cases with treatment	311,443	143,394	144,508	145,446
Cervical cancer deaths	282,687	130,159	131,170	132,021
DALYs (discounted *)	1,695,103	786,204	792,228	797,304
ECONOMIC OUTCOMES				
Healthcare treatment costs (USD)	65,657,026	30,464,253	30,697,492	30,894,019
Vaccination programme cost (USD)				
Discounted (3%)	-	37,450,569	37,450,569	37,581,339
No discount	-	42,074,184	42,074,184	42,224,497
Cost (USD) per DALY averted (compared to no vaccine) *				
Cost	-	2,257,796	2,491,034	2,818,332
DALYs averted	-	908,898	902,875	897,799
Cost per DALY averted (with Gavi support)	-	2.5	2.8	3.1
Cost (USD) per DALY averted * (compared to next least costly non-dominated ** option)				
Cost	-	2,257,796	Dominated **	Dominated **
DALYs averted	-	908,898	Dominated **	Dominated **
Cost per DALY averted	-	2.5	Dominated **	Dominated **

* Future costs/effects were discounted at a rate of 3% per year. ** A product is dominated if at least one other product provides greater benefits at lower cost.

**Table 6 vaccines-11-01058-t006:** Lifetime effects and costs of vaccinating 14 cohorts of preadolescent girls over the period 2022–2031 in Mozambique (with Gavi support, with cross-protection).

OUTCOMES	No Vaccine	CERVARIX	CECOLIN	GARDASIL-4
HEALTH OUTCOMES				
Cervical cancer cases (local)	63,637	20,187	28,828	29,527
Cervical cancer cases (regional)	249,451	79,132	113,004	115,744
Cervical cancer cases (distant)	29,158	9,250	13,209	13,529
Cervical cancer cases with treatment	311,443	98,798	141,088	144,508
Cervical cancer deaths	282,687	89,685	128,065	131,170
DALYs (discounted *)	1,695,103	550,289	773,729	792,228
ECONOMIC OUTCOMES				
Healthcare treatment costs (USD)	65,657,026	21,342,264	29,981,215	30,697,492
Vaccination program cost (USD)				
Discounted (3%)	-	37,581,339	37,450,569	37,450,569
Undiscounted	-	42,074,184	42,074,184	42,224,497
Cost (USD) per DALY averted (compared to no vaccine) *				
Cost	-	−8,273,533	1,774,758	2,491,034
DALYs averted	-	1,184,261	921,373	902,875
Cost per DALY averted	-	Cost saving	1.9	2.8
Cost (USD) per DALY averted * (compared to next least costly non-dominated ** option)				
Cost	-	−8,273,533	Dominated **	Dominated **
DALYs averted	-	1,184,261	Dominated **	Dominated **
Cost per DALY averted	-	Cost saving	Dominated **	Dominated **

* Future costs/effects were discounted at a rate of 3% per year. ** A product is dominated if at least one other product provides greater benefits at lower cost.

**Table 7 vaccines-11-01058-t007:** Lifetime effects and costs of vaccinating 14 cohorts of preadolescent girls over the period 2022–2031 in Mozambique (without Gavi support, without cross-protection).

OUTCOMES	No Vaccine	CECOLIN	GARDASIL-4	CERVARIX
HEALTH OUTCOMES				
Cervical cancer cases (local)	63,637	29,299	29,896	29,719
Cervical cancer cases (regional)	249,451	114,852	117,190	116,496
Cervical cancer cases (distant)	29,158	13,425	13,698	13,617
Cervical cancer cases with treatment	311,443	143,394	146,314	145,446
Cervical cancer deaths	282,687	130,159	132,809	132,021
DALYs (discounted *)	1,695,103	786,204	801,961	797,304
ECONOMIC OUTCOMES				
Healthcare treatment costs (USD)	65,657,026	30,464,253	31,074,244	30,894,019
Vaccination programme cost (USD)				
Discounted (3%)	-	59,745,515	72,957,335	80,987,673
No discount	-	68,017,900	83,391,954	92,734,670
Cost (USD) per DALY averted (compared to no vaccine) *				
Cost	-	24,552,742	38,374,553	46,224,666
DALYs averted	-	908,898	893,142	897,799
Cost per DALY averted (with Gavi support)	-	27	43	52
Cost (USD) per DALY averted * (compared to next least costly non-dominated ** option)				
Cost	-	24,552,742	Dominated **	Dominated **
DALYs averted	-	908,898	Dominated **	Dominated **
Cost per DALY averted	-	27	Dominated **	Dominated **

* Future costs/effects were discounted at a rate of 3% per year. ** A product is dominated if at least one other product provides greater benefits at less cost.

**Table 8 vaccines-11-01058-t008:** Lifetime effects and costs of vaccinating 14 cohorts of preadolescent girls over the period 2022–2031 in Mozambique (without Gavi support, with cross-protection).

OUTCOMES	No Vaccine	CECOLIN	CERVARIX	GARDASIL-4
HEALTH OUTCOMES				
Cervical cancer cases (local)	63,637	28,828	20,187	29,527
Cervical cancer cases (regional)	249,451	113,004	79,132	115,744
Cervical cancer cases (distant)	29,158	13,209	9,250	13,529
Cervical cancer cases with treatment	311,443	141,088	98,798	144,508
Cervical cancer deaths	282,687	128,065	89,685	131,170
DALYs (discounted *)	1,695,103	773,729	550,289	792,228
ECONOMIC OUTCOMES				
Healthcare treatment costs (USD)	65,657,026	68,017,900	89,421,090	83,391,954
Vaccination program cost (USD)				
Discounted (3%)	-	59,745,515	80,987,673	72,957,335
No discount	-	68,017,900	92,734,670	83,391,954
Cost (USD) per DALY averted (compared to no vaccine) *				
Cost	-	24,069,704	36,672,910	37,997,801
DALYs averted	-	921,373	1,144,814	902,875
Cost per DALY averted	-	26	32	42
Cost (USD) per DALY averted * (compared to next least costly non-dominated ** option)				
Cost	-	24,069,704	12,603,206	Dominated **
DALYs averted	-	921,373	223,440	Dominated **
Cost per DALY averted	-	26	6	Dominated **

* Future costs/effects were discounted at a rate of 3% per year. ** A product is dominated if at least one other product provides greater benefits at lower cost.

## Data Availability

All the data used in this study was obtained from open platforms GLOBOCAN, UNIPOP, GAVI, manuscripts and from meetings/conversations with experts.

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
