# Peer review of "Impact and Cost-Effectiveness of Alternative Human Papillomavirus Vaccines for Preadolescent Girls in Mozambique: A Modelling Study"

_vaccines, 2023, doi:10.3390/vaccines11061058_

Round 1
Reviewer 1 Report
Paper is well written and nicely prsented with sufficient facts and figures.
Reviewer 2 Report
If I understand the thrust of this report, the authors sought to evaluate 1) the cost-effectiveness of a vaccine program in terms of the overall public health burden of HPV-associated disease. 2) projecting the efficacy of various HPV vaccine formulations.
The authors do a fairly good job in making this case; although specific details on data acquisition to support their findings was not very specific.
My major concern with this manuscript is with respect to the conclusion that they arrive after this article: "Based on available evidence about cross-protection 352 CERVARIXTM could have more impact than both CECOLIN and GARDASIL-4 and is worth consideration".
In the manuscript, the authors do not demonstrate sufficient clinical or basic science to support this recommendation for this particular population. Thus, this reviewer doesn't feel this claim is justified. It is concerning since the take-home message doesn't speak to the public health relevancy of costs relative to the disease burden of HPV vaccination programs, which is truly what is described in this article. The bottom line, the data presented here do not support the conclusions.
Reviewer 3 Report
This is a well-written article to investigate the cost-effectiveness of three HPV vaccines for preadolescent girls in Mozambique but I have the following concerns.
1. Why did the authors estimate the efficacy of one dose instead of the two doses required?
2. Would it be possible for the authors to present a cost-effectiveness frontier instead of presenting the "Cost (US$) per DALY averted* (compared to next least costly nondominated** option)" in the tables?
3. I've noticed that in the complementary file Tables 1 and 2, when the healthcare costs were varied between low and high, the cost per DALY of Cecolin* with Gavi support was lower when the healthcare costs was low than when the healthcare costs was high but this wasn't the case for the other interventions as well as when there were no Gavi support. Likewise, this trend is observed for Gardasil-4** when the healthcare costs were high and it having higher cost per DALY without Gavi support than when the healthcare costs were low.
why is there a difference in direction?
Cecolin* Cervarix Gardasil-4
(a)<(b) (a)>(b) (a)>(b)
where
(a) Healthcare costs = Low with Gavi support
(b) Healthcare costs = High with Gavi support
Cecolin Cervarix Gardasil-4**
(c)>(d) (c)>(d) (c)<(d)
where
(c) Healthcare costs = Low without Gavi support
(d) Healthcare costs = High without Gavi support
4. Figure 1, the top two graphs does not contain the willingness-to-pay threshold, can it be added to make it consistent please?
5. What's complementary file Figure 1 for?
Round 2
Reviewer 3 Report
All concerns have been adequately addressed.
Author Response
The authors thank the reviewer for the information on the status of the responses provided to the first round of reviewer comments/suggestions.